# Peripheral Blood Levels of Brain-Derived Neurotrophic Factor in Patients with First Episode Psychosis: A Systematic Review and Meta-Analysis

**DOI:** 10.3390/brainsci12040414

**Published:** 2022-03-22

**Authors:** Sukhbir Singh, Dipta Roy, Taylor Marzouk, Jian-Ping Zhang

**Affiliations:** 1Division of Psychiatry Research, Zucker Hillside Hospital, Northwell Health, Glen Oaks, NY 11004, USA; ssingh4612@gmail.com (S.S.); dipta.roy52@myhunter.cuny.edu (D.R.); tmarzouk@northwell.edu (T.M.); 2Advarra Inc., Institutional Review Board, Columbia, MD 21044, USA; 3Department of Psychiatry, Weill Cornell Medical College, White Plains, NY 10605, USA

**Keywords:** BDNF, first episode psychosis, antipsychotic treatment

## Abstract

Background: Peripheral blood level of brain-derived neurotrophic factor (BDNF) may be used as a diagnostic and/or prognostic marker for schizophrenia. Previous studies were inconsistent. A systematic review was conducted to examine whether BDNF level is different in patients with first episode psychosis (FEP) compared to health controls (HC) and whether it changes after treatment. Methods: Literature search was done in PubMed, Web of Science, and Google Scholar following standard procedures. Hedges’ g was used as the measure of effect size (ES), which was pooled with random effects model. Publication bias and moderator effects were examined. Results: Search yielded 29 studies with a total sample size of 2912. First meta-analysis included 27 studies with FEP vs. HC comparison. Pooled ES was −0.63, *p* < 0.001, indicating that BDNF level was lower in FEP than in HC. Studies were heterogeneous, and moderator analysis showed that studies of younger patient, higher symptom severity, and more drug naïve had larger ES. Second meta-analysis examined change in BDNF levels before and after antipsychotic treatment in eight studies. A pooled ES of −0.003 (*p* = 0.96) showed no change in peripheral BDNF level after treatment. Conclusion: Peripheral BDNF level was decreased in FEP compared to HC, but it did not change after treatment.

## 1. Introduction

Schizophrenia is a neuropsychiatric disorder that affects 1% of the world population [1]. This debilitating disorder is characterized by deficits in thought processing, emotional responsiveness, and neurocognition [2]. Although the pathophysiology of this disease is still unclear, it has been speculated that dysregulation of neurotrophic factors may play an important role [3]. These factors include nerve growth factor, brain-derived neurotrophic factor, and other neurotrophins [4,5].

Brain-Derived Neurotrophic Factor (BDNF) promotes growth, differentiation, and survival of nerve cells [5,6]. BDNF exists in two phases: its precursor and its mature form and binds to p75 and tropomyosin receptor kinase B (TrkB) receptors, respectively [7,8]. BDNF has been shown to promote the survival of dopamine neurons in culture and promotes the formation of the D1 receptor [9]. Studies have shown that BDNF levels are decreased in the prefrontal cortex and hippocampus in patients with schizophrenia [10]. In addition, high levels of dopamine can block transcription of the BDNF gene [9,11,12]. Low levels of BDNF also causes a reduction of tyrosine hydroxylase, which leads to neuronal dysfunction [13]. In rodent models, there is correlation between drug induced psychoses with lower levels of BDNF mRNA [14,15].

Targeting dopamine pathways by antipsychotic drugs is the mainstay therapy for schizophrenia. Therefore, it is plausible that the utilization of antipsychotic drugs could lead to an upregulation of BDNF expression. It has been shown that BDNF has the ability to cross the blood–brain barrier [16], and it is readily measured in the peripheral blood [17,18]. As such, this could provide an opportunity to uncover and monitor the possible pathophysiological changes that are occurring in the brain of a psychotic patient.

BDNF could serve as a potential diagnostic and prognostic biomarker, which may allow us to detect schizophrenia early and track treatment response. However, available data on peripheral levels of BDNF in psychotic patients compared to healthy controls is not consistent [19]. For example, two studies showed a decrease of BDNF in a sample of psychotic patients compared to healthy controls [20,21], while other studies showed no statistical difference between patients and healthy controls [22,23]. Yet, some studies have also shown that peripheral BDNF levels are increased in patients versus healthy controls [24]. One potential cause of the inconsistency is that some studies used first episode patients but others used chronic patients. Previous research has investigated the effect of antipsychotic treatment on peripheral BDNF levels in longitudinal studies, but the data are also inconsistent. One study showed an increase of peripheral BDNF levels from before to after antipsychotic treatment [22]. In contrast, two later studies reported a decrease or no increase in peripheral BDNF levels after treatment [24,25]. Again, chronicity of the patient samples may explain the discrepant findings. Meta-analysis on this topic has been attempted to reconcile the differences among studies. For example, a meta-analytic study published in 2015 showed a decrease of both drug naive and medicated schizophrenia patients peripheral BDNF levels when compared to healthy controls [26]. Since its publication, there has been additional studies reporting data on BDNF levels in schizophrenia.

The aim of the present study was to systematically review the literature on this topic and meta-analyze the data on peripheral BDNF levels of first episode psychotic (FEP) patients compared with aged matched healthy controls (HC). The analysis focused on the first episode patient samples at the time of initial onset of symptoms when most patients were antipsychotic drug naïve or had minimal exposure to antipsychotics. This would avoid potential confounding effects of receiving chronic treatment on peripheral BDNF levels. The present review also examined studies that provided longitudinal data after the start of antipsychotic treatment to assess the validity of BDNF being a prognostic tool. We hypothesized that peripheral BDNF levels would be lower in FEP when compared to healthy controls, and that BDNF would increase after receiving antipsychotic treatment, potentially reflecting treatment response.

## 2. Methods

### 2.1. Literature Search

Literature search was conducted up to 30 June 2021. The following databases were used: PubMed, Web of Sciences, and Google Scholar. Search was done using the following keywords and search terms: “(BDNF OR (Brain-Derived Neurotrophic Factor)) AND schizophrenia”, (BDNF OR (Brain-Derived Neurotrophic Factor)) AND (first episode psychosis), and (BDNF OR (Brain-Derived Neurotrophic Factor)) AND schizophrenia AND (Drug naïve)”. Abstracts of each paper were carefully inspected first. If an abstract appeared eligible, full text was reviewed to determine eligibility. Additionally, cited references in review papers on BDNF and first episode psychosis were retrieved and reviewed.

### 2.2. Selection Criteria

In this meta-analysis, we included studies restricted to human studies using the following inclusion criteria: (1) published in a peer-reviewed journal; (2) reported data on BDNF levels; (3) included patients with first episode psychosis who met the ICD-10, DSM-III-R, DSM-IV, or DSM-V criteria for schizophrenia, schizoaffective disorder, psychotic disorder NOS, and schizophreniform disorder; and (4) had age-matched healthy controls, or had longitudinal follow-up data after antipsychotic treatment. Criteria for exclusion were as follows: (1) included patients mostly with no psychotic spectrum disorders, (2) no clearly defined patient group, (3) most patients had chronic schizophrenia, and (4) did not report detailed data to calculate effect sizes. The authors of studies were contacted for more information or clarification if needed.

### 2.3. Data Extraction

The following variables were extracted: First author name, publication year, race of patient sample, mean age, age of onset, duration of illness, medications, percentage of patients who were drug naïve, peripheral blood BDNF levels, source of BDNF levels (serum or plasma), and psychotic symptom scores via rating scales. The extraction of this data was conducted by authors SS and DR. If there was any discrepancy, JZ would verify and resolve the differences by discussion among the research team.

### 2.4. Statistical Analysis

Meta-analyses were conducted by using the Comprehensive Meta-Analysis (CMA) software version 2.0 (Biostat, Eaglewood, NJ, USA). Since the studies varied in their measurement methods, standardized mean differences were calculated for the BDNF levels of the first episode psychotic subjects in comparison with healthy controls. Hedges’ g was used as the effect size measure to assess the difference between groups, or the difference before and after treatment in a longitudinal study. A pooled effect size (ES) was computed with a random effects model. Heterogeneity among studies was assessed by using the Q test and *I*^2^. The level of heterogeneity is categorized by low, medium and high by a *I*^2^ measurement of <25%, ~50%, and >75%, respectively [27]. If high heterogeneity was found, moderator analysis and meta-regression were conducted to examine the sources of heterogeneity. We also assessed publication bias by utilizing Egger’s regression test, Classic Fail-Safe N, and the “Trim and Fill” method [28]. To assess the influence of potential outliers on the pooled results, pooled effect sizes were computed by removing one study at a time to examine whether the pooled effect changes significantly.

## 3. Results

### 3.1. Search Results

Figure 1 summarized the literature search process by using the Preferred Reporting Items for Systematic Reviews and Meta-Analyses (PRISMA) flow chart. A total of 1444 publications were screened, out of which 1399 were excluded by abstract and 45 full-text articles were assessed. At the end, 29 studies were included in the meta-analysis. Relevant descriptive and demographic data from each study is shown in Table 1, which also included summary data of BDNF and symptom rating scale scores from each study.

The total sample size of included studies was 2912. Twenty-seven out of the 29 studies reported baseline BDNF peripheral blood levels in both FEP and HCs. Among these studies, 10 and 17 utilized plasma and serum peripheral BDNF levels, respectively. The mean ages of FEP and HC were 27.35 and 29.21 years old, respectively. The locations of these studies consisted of 13 in Asia, 14 in Europe, and two in North America. A total of 17 out of the 29 studies included 100% patients who were antipsychotic drug naive. The percentage of male patients in these studies ranged from 30 to 70%. A total of 10 studies reported peripheral BDNF levels before and after antipsychotic treatment, and eight were included in the analysis. These follow up studies all included a baseline and at least one follow-up time point of peripheral BDNF levels. Two studies [34,43] were excluded due to lack of *t*-test values or p-values needed to calculate effect size for the change of BDNF before and after antipsychotic treatment.

### 3.2. Baseline BDNF Levels in FEP versus HC 

A total of 27 studies with a pooled sample size of 2524 reported baseline peripheral BDNF levels in both FEP and HC, so they were included in the meta-analysis. The pooled ES was −0.63 (95% CI: −0.89 to −0.37, *p* < 0.001) (Figure 2). FEP had significantly lower peripheral BDNF levels when compared to age-matched HC. Heterogeneity across studies was high, Q = 240.6, df = 26, *p* < 0.001, *I*^2^ = 89.6%. Therefore, moderator analysis and meta-regression were performed. The moderator analysis showed that studies with larger sample sizes tended to have smaller ES, suggesting less difference in BDNF levels between FEP and HC (*p* = 0.04). In addition, studies that contained higher percentage of male subjects tended to have smaller ES. In contrast, studies with lower mean ages, more drug naive subjects, and subjects with higher symptom severity tended to report lower BDNF levels in FEP compared to HC (*p*’s < 0.05). There was no significant moderation effect of publication year, study regions (Asia versus Europe versus North America, Q = 1.29, df = 2, *p* = 0.53), duration of untreated psychosis, or ethnicity in the samples. Table 2 showed results of meta-regression. A subgroup analysis was also done to see whether there was a difference between studies that evaluated serum samples versus those using plasma levels. The subgroup analysis showed no significant difference between plasma and serum peripheral blood samples. 

The publication bias was assessed by Egger’s regression test, the classic Fail-Safe number, and the “Trim and Fill Method”. The Egger’s regression test did not detect publication bias, beta = −3.43, *t* = 1.78, *p* = 0.09 (2 tailed). The classic Fail-Safe number was 1180, indicating that in order to make the pooled effect size non-significant, we would need over 1000 unpublished studies favoring FEP, which is unlikely. The “Trim and Fill Method” yielded six missing studies that favored the healthy controls. These six missing or unpublished studies would have shown even lower BDNF levels in FEP when compared to HC. If the missing studies were added to the analysis the pooled ES would have been larger, around −0.90. Therefore, potential publication bias would not alter the results towards the null finding.

### 3.3. Change in BDNF Levels in FEP before and after Treatment

The second meta-analysis included eight studies with a total sample size of 241 that reported peripheral BDNF levels in FEP before and after antipsychotic treatment. The pooled ES was 0.003 (*p* = 0.96) (Figure 3). There was no significant change in BDNF levels before and after antipsychotic treatment. This indicates that as FEP subjects receive treatment there is no increase of peripheral BDNF levels. There was no significant heterogeneity across studies (Q = 5.97, *p* = 0.54, *I*^2^ = 0%), and the duration of follow-up did not significantly moderate the ES. 

The publication bias was assessed by Egger’s regression test and the “Trim and Fill Method”. The Egger’s regression test did not detect publication bias, beta = −0.43, *t* = 0.23, *p* = 0.82 (2 tailed). The “Trim and Fill Method” yielded one missing study that favored higher baseline. However, the one missing or unpublished study would not have changed the pooled ES significantly. The adjusted pooled ES was −0.01, *p* > 0.05. The Fail-Safe number was not reportable because the pooled ES was not statistically significant.

## 4. Discussion

The goal of the present study was to assess the plausibility of using peripheral BDNF levels as a marker for diagnosis of schizophrenia and treatment response to antipsychotic drug treatment. The first meta-analysis examined 27 studies that compared baseline BDNF levels between FEP subjects and age-matched HC. The result showed robust difference between the two groups with BDNF levels significantly lower in FEP. The second analysis focused on eight studies that assessed the BDNF levels in FEP longitudinally did not find significant change from baseline to after antipsychotic treatment. To our knowledge, this is the largest meta-analysis on this topic that focuses on patients with first episode schizophrenia.

In 27 studies with a total sample size of 2524 subjects, peripheral BDNF levels in FEP were significantly lower than those in HC, with a pooled ES of −0.63, a moderate to large effect size. This is consistent with previous studies [26,52]. The meta-analysis published in 2015 [26] examined both studies of first episode and chronic patients, and 17 first episode studies were included. Our study found another 10 papers published after 2015. Although there are discrepancies in effect size calculation, the conclusions are similar. Therefore, the finding that peripheral blood BDNF levels are lower in patients with first episode psychosis seems robust. Previous studies also found that multi-episode patients also had lower BDNF levels when they were acutely psychotic [26]. In the present study, patients with FEP had less peripheral BDNF circulating in their blood compared to the healthy controls. The substantial heterogeneity was partially moderated by several actors; effect sizes were larger in samples that were younger, more drug naïve and higher symptom severity. The effects of age is consistent with the results from a previous meta-analysis by Green and colleagues [52], which found that the BDNF concentration decreases as the age increased.

Although it may be promising to use peripheral BDNF level as a diagnostic biomarker for psychosis, the specificity of this marker remains unclear. Previous research has found that peripheral BDNF levels were also reduced in patients with major depressive disorder and bipolar affective disorder who were acutely depressed [53]. As such, reduced BDNF levels appeared to cut cross different psychiatric diagnostic categories and is not specific to psychosis or schizophrenia. Perhaps the elevated BDNF levels represent more of high stress levels [54] associated with the acute phase of psychosis and depression, rather than indicating a specific pathophysiological pathway associated with schizophrenia. Glucocorticoid secretion related to stress is linked to altered neuronal functionality [55]. Studies have demonstrated that under stressful conditions, BDNF is decreased in different areas of the brain and that glucocorticoid administration caused a decrease of hippocampal BDNF concentration [56]. The relationship between stress and decreased levels of BDNF will need further investigation.

When looking at low levels of peripheral BDNF in FEP, it has been demonstrated that different areas of the brain in patient with schizophrenia have decreased concentrations of BDNF [10,57,58]. The decrease of BDNF is possibly related to a decrease of synthesis [10]. It is hypothesized that the decrease of peripheral BDNF is a direct result of the decrease centrally because it has been demonstrated that BDNF crosses the blood brain barrier. What is the cause of the decrease of BDNF centrally and in the periphery? Our meta-regression showed that BDNF levels were decreased in patients with more severe symptoms, although we cannot make conclusions on the directionality of this correlation. An epigenetic study also reported a correlation between major psychosis and DNA methylation of the BDNF gene [59]. Further research is required focusing solely on psychotic symptoms with their relationship to BDNF.

Whether peripheral BDNF levels can be used as a biomarker for antipsychotic treatment response was examined in the second meta-analysis on BDNF levels before and after treatment. If BDNF levels can serve as a prognostic tool, it must change with treatment. Ideally, we would be able to see that BDNF levels increase in treatment responders but no change in treatment non-responders. However, most studies did not report BDNF levels in treatment responders and non-responders separately. Given the fact that most patients with first episode psychosis respond well to antipsychotic treatment, we should be able to see change in BDNF levels in the whole sample if it were predictive of treatment. In eight studies with a total sample size of 241 FEP patients, the pooled effects size is 0.003, not statistically significant. This is different from what a previous meta-analysis found [26], which included 11 studies, but some were on multi-episode patients. Our analysis is strictly based on studies with FEP, and five studies were published after the previous meta-analysis. It seems that among FEP patients who were antipsychotic drug naïve or had minimal drug exposure, BDNF did not significantly increase after antipsychotic treatment. Therefore, unfortunately, there is no strong evidence that peripheral BDNF levels can be used as a biomarker for treatment response.

Another perspective that needs to be taken into consideration when examining the decrease of peripheral levels of BDNF is neuroinflammation. There has been a suggested link between inflammation and immune dysfunction with the pathogenesis of schizophrenia [60,61,62]. There is also an inverse correlation between inflammatory markers and BDNF levels specifically with IL-6 and BDNF mRNA levels; as IL-6 increases BDNF mRNA decreases, and vice versa [54,61,63]. Furthermore, it has been hypothesized that a central component of decrease in formation of BDNF is methylation of the BDNF gene directly caused by the gene interaction with proinflammatory markers [61].

There are a few limitations of the present meta-analytic review. First, most studies were not designed to test the hypothesis that BDNF levels may help schizophrenia diagnosis or predict treatment response. The reported findings on BDNF were from secondary data analysis, which might increase type I error rate, resulting in publication bias. Second, many studies had small sample sizes. Average sample size among the studies included in the meta-analysis was less than 95. Finally, the number of longitudinal studies was small, and it was not possible to separate BDNF levels for treatment responders and non-responders. As such, the test for whether BDNF levels can serve as a biomarker for antipsychotic treatment is only preliminary.

## 5. Conclusions

The present meta-analyses showed that peripheral BDNF levels are decreased in patients with FEP when compared to healthy controls. As the heterogeneity across studies was large, our moderator analysis showed that patients that were younger, more drug naïve, and higher symptom severity tended to have much lower BDNF levels compared to the healthy controls. When comparing FEP subjects before and after antipsychotic treatment, there were no significant change in the BDNF levels. Additional robust and longitudinal studies need to be conducted to assess the clinical correlates of BDNF alterations. Therefore, there is no strong evidence to support the idea that peripheral BDNF levels can be used as a diagnostic marker or marker for treatment response.

## Figures and Tables

**Figure 1 brainsci-12-00414-f001:**
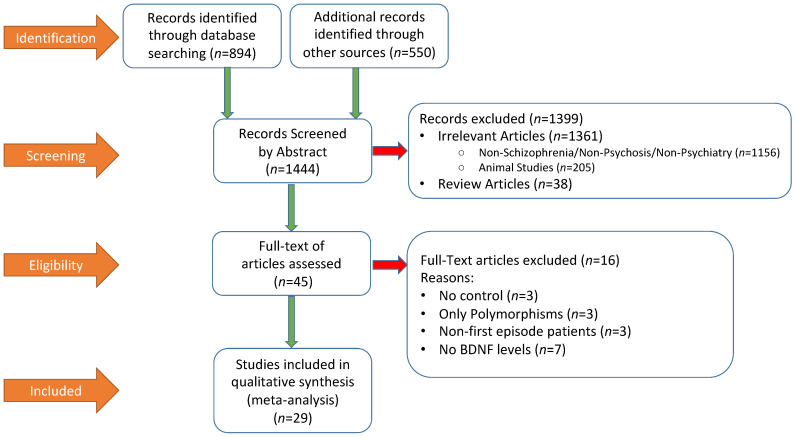
Preferred Reporting Items for Systematic Reviews and Meta-Analyses (PRISMA) flow diagram of literature search procedure and results.

**Figure 2 brainsci-12-00414-f002:**
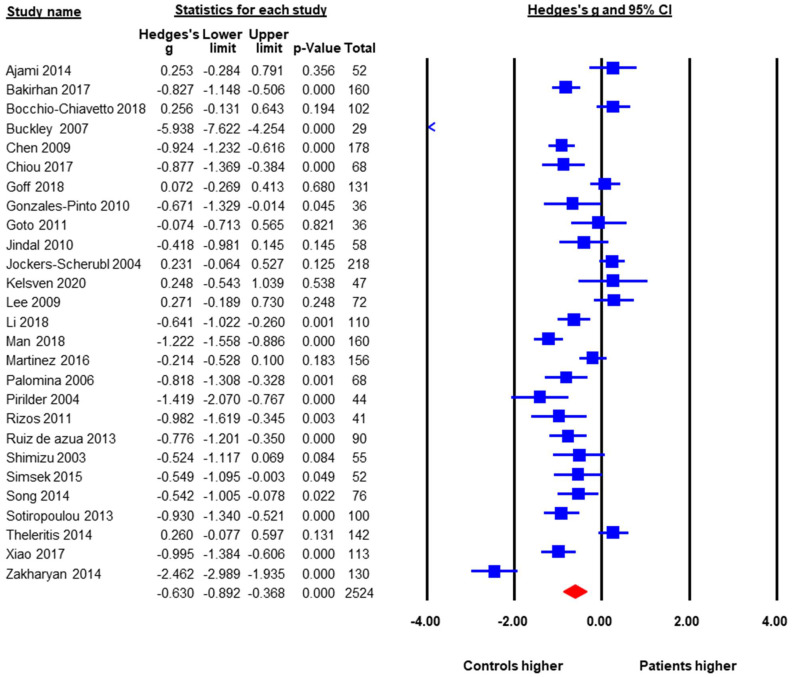
Peripheral BDNF Levels in First Episode Psychotic Patients versus Healthy Controls.

**Figure 3 brainsci-12-00414-f003:**
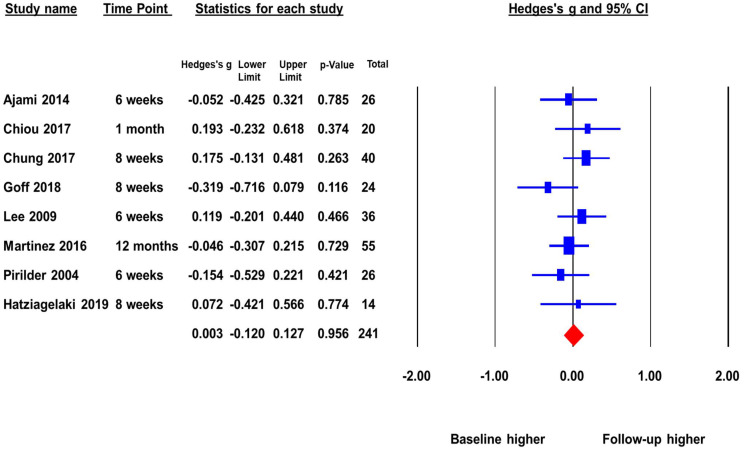
Peripheral BDNF levels in first episode psychotic patients before and after antipsychotic treatment.

**Table 1 brainsci-12-00414-t001:** Demographical data of included studies.

Author (Year)	Location	FU Duration	Dx Crit.	Total N	Subjects	Age ± SD	% Male	% Cauc	% Drug Naïve	BDNF Source	Total BDNF (BL & FU): Mean ± SD	Clinical Ratings (BL & FU): Mean ± SD
Ajami A. (2014) [24]	Asia	40 Days	DSM IV	52	FE HC	33.61 ± 9.49 33.92 ± 8.86	---	---	---	Serum	FE BL: 4.128 ± 4.017 FE FU: 4.065 ± 3.648 HC BL: 3.18 ± 3.32	BL: PANSS-T: 87.61 ± 22.99 PANSS-N: 25.46 ± 7.77 FU: PANSS-T: 52.93 ± 12.63 PANSS-N: 15.92 ± 6.79
Bakirhan (2017) [29]	Asia	---	DSM IV	160	DN SZ HC	31.08 ± 9.37 29.65 ± 8.78	58.8% 58.8%	---	100%	Serum	DN: 18.75 ± 5.62 HC: 22.35 ± 2.44	PANSS-T: 110.63 ± 9.76 PANSS-P: 28.92 ± 3.37 PANSS-GP: 51.25 ± 4.29 PANSS-N: 30.42 ± 4.10
Bocchio-Chiavetto (2018) [30]	Europe	9 months	ICD-10	102	FE HC	28.7 ± 8.6 32.0 ± 5.1	64.2% 51%	---	---	Serum	FE: 42.7 ± 9.0 HC: 40.3 ± 9.6	PANSS-T: 2.60 ± 0.62 PANSS-P: 2.43 ± 0.79 PANSS-G: 2.53 ± 0.53 PANSS-N: 2.94 ± 1.20
Buckley (2007) [20]	USA	---	DSM IV	29	FE HC	21.8 ± 8.83 25.28 ± 5.72	64% 53%	NA	100%	Plasma	FE: 17 ± 11.22 HC: 49.1 ± 25.95	---
Chen (2009) [31]	Asia	---	DSM IV	178	FE HC	29.2 ± 9.6 29.9 ± 9.8	53% 54%	0%	100%	Serum	FE: 9.0 ± 4.2 HC: 12.1 ± 2.2	PANSS-P: 25.6 ± 6.3 PANSS-N: 18.3 ± 7.2 PANSS-GP: 39.6 ± 11.5 PANSS-T: 83.4 ± 18.9
Chiou (2017) [25]	Asia	4 weeks	DSM IV	BL (68) FU (20)	DN FE HC	30.6 ± 11.2 30.7 ± 8.1	35.3% 35.3%	---	100%	Serum	FE BL: 9.4 ± 6.8 (n = 68) HC BL: 15.3 ± 6.5 (n = 20) FE BL: 9.2 ± 7.8 FE FU: 10.1 ± 6.2	PANSS-P: 28.1 ± 5.6 PANSS-N: 26.6 ± 4.9 PANSS-GP: 58.1 ± 6.1 PANSS-T: 121.3 ± 13.7
Chung (2017) [32]	Asia	8 weeks	DSM IV	BL (75) FU (51)	FE	30.8 ± 11.0	54.7%	---	58.7%	Plasma	FE BL:2848.00 ± 1599.00 (n = 61) FE FU: 2873.00 ± 1509.00 (n = 42)	BL PANSS-T: 99.55 ± 20.25 (n = 75) FU PANSS-T: 65.88 ± 19.23 (n = 51)
Goff (2018) [33]	Asia	8 weeks	DSM IV	144	FE HC	25.2 ± 7.7 23.9 ± 6.4	45.1% 45.2%	---	100%	Plasma	FE BL: 926.81 ± 1351.55 (n = 69) HC: 807.49 ± 1937.15 (n = 62) FE BL: 1399.95 ± 1790.20 (n = 24) FE FU: 749.54 ± 796.65 (n = 24)	BPRS-T: 48.01 ± 10.97 BPRS-P: 19.11 ± 4.80 BPRS-N: 5.81 ± 3.27 SANS: 20.74 ± 17.95
Gonzalez-Pinto (2010) [34]	Europe	1 year	DSM IV	36	DN FE HC	24.39 ± 6.53 25.19 ± 5.95	---	---	100%	Plasma	FE BL: 4.09 ± 2.31 HC BL: 5.80 ± 2.66	PANSS-P: 24.4 ± 4.8 PANSS-GP: 40.94 ± 13.95 PANSS-N: 13.22 ± 6.96
Goto (2011) [35]	Asia	---	DSM IV	36	FE HC	29 ± 11 30 ± 11	50% 50%	---	100%	Serum	---	PANSS-P: 15.9 ± 4.2 PANSS-N: 17.2 ± 5.3 PANSS-G: 34.2 ± 10.1 PANSS-T: 68.1 ± 17.0
Hatziagelaki (2019) [36]	Europe	8 weeks	DSM IV	14	DN FE	26.50 ± 6.02	50%		100%	Serum	FE BL: 12.98 ± 57.74 FE FU: 17.18 ± 45.73	FE BL-PANSS-P: 40.28 ± 5.19 PANSS-N: 29.85 ± 5.43 FE FU- PANSS-P: 25.92 ± 5.35 PANSS-N: 23.14 ± 5.06
Jindal (2010) [37]	North America	6 months	DSM IV	82	DN FE HC	22.4 ± 5.47 22.31 ± 5.67	70.8% 61.0%	---	100%	Serum	FE: 97.58 ± 31.41 HC: 116.78 ± 38.42	SAPS: 24.29 ± 10.57 SANS: 41.7 ± 10.41
Jockers-Scherubl (2004) [38]	Europe	---	DSM IV	229	DN FE HC	31.8 32.3	57.3% 45.9%	---	100%	Serum	DN FE: 13.1 ± 5.9 HC: 13.2 ± 5.2	---
Kelsven (2020) [39]	North America	---	DSM-IV	57	FE HC	24.06 ± 6.14 20.40 ± 4.90	76.0% 71.4%	72.0% 71.4%	80.0%	Plasma	FE: 5015.95 ± 6038.56 HC: 3875.10 ± 1165.39	PANSS-P: 25.08 ± 10.00 PANSS-N: 21.22 ± 9.82 PANSS-G: 45.58 ± 15.08 PANSS-T: 91.8 ± 31.96
Lee (2009) [22]	Asia	6 weeks	DSM IV	72	DN FE HC	31.3 ± 7.8 31.3 ± 7.9	41.7% 41.7%	---	100%	Plasma	BL FE: 980.08 ± 452.48 FU FE: 1030.95 ± 546.32 HC: 880.61 ± 244.57	BL: PANSS-P: 26.4 ± 7.0 PANSS-N: 22.7 ± 9.3 PANSS-G: 47.4 ± 12.2 FU:PANSS-P: 14.3 ± 6.3 PANSS-N: 15.7 ± 8.4 PANSS-G: 28.3 ± 10.8
Li (2018) [40]	Asia	---	DSM V	110	DN FE HC	28.15 ± 10.42 31.33 ± 10.69	50.9% 50.9%	---	100%	Serum	FE: 24.30 HC: 35.30	PANSS-T: 75.81 ± 21.35 PANSS-P: 16.58 ± 7.47 PANSS-N: 18.93 ± 7.83
Man (2018) [41]	Asia	3 months	DSM IV	160	FEDN HC	25.7 ± 8.9 34.9 ± 8.8	53.8% 57.5%	0%	100%	Serum	DN FE: 8.8 ± 3.1 HC: 12.1 ± 2.2	PANSS-P: 19.7 ± 5.7 PANSS-N: 18.3 ± 7.1 PANSS-G: 33.1 ± 8.4 PANSS-T: 70.8 ± 15.1
Martinez (2016) [42]	Europe	12 months	DSM IV	174	DN FE HC	23.78 ± 5.81 25.69 ± 7.07	69.1% 66.3%	---	100%	Plasma	---	PANSS-P: 10.21 ± 4.99 PANSS-N: 14.11 ± 5.73 PANSS-T: 50.44 ± 16.89
Palomino (2006) [43]	Europe	12 months	DSM IV	91	DN FE HC	23.7 ± 1 25.5 ± 0.8	---	---	100%	Plasma	FE BL: 4.66 ± 2.61 HC: 7.55 ± 4.31	---
Pirildar (2004) [21]	Europe	6 weeks	DSM IV	44	FE HC	27.81 ± 9.54 25.7 ± 5.8	31.8% 31.8%	---	22.7%	Serum	BL FE: 14.19 ± 8.12 FU FE: 14.53 ± 2.93 BL HC: 26.8 ± 9.3	BL:PANSS-P: 30.81 ± 7.99 PANSS-N: 30.27 ± 7.31 PANSS-G: 58.04 ± 11.32 FU: PANSS-P: 16.45 ± 8.96 PANSS-N: 17.63 ± 11.32 PANSS-G: 33.57 ± 15.03
Rizos (2011) [44]	Europe	---	DSM IV	41	DN FE HC	30.75 ± 10.52 34 ± 4.70	40% 52.4%	---	100%	Serum	FE: 9.76 ± 4.61 HC: 15.33 ± 6.34	PANSS-P: 30.94 ± 4.30 PANSS-N: 31.66 ± 9.07
Ruiz de Azua (2013) [45]	Europe	6 months	DSM IV	90	FE HC	24.3 ± 8.5 24.0 ± 8.8	55.6%	100%	88.8%	Plasma	FE: 6.09 ± 3.70 HC: 9.19 ± 4.21	---
Shimizu (2003) [23]	Asia	---	DSM IV	80	FE HC	35.6 ± 14.1 36.5 ± 14.6	50% 50%	---	100%	Serum	DN FE: 23.8 ± 8.1 HC: 28.5 ± 9.1	BPRS-P: 18.0 ± 10.2 BPRS-N: 4.60 ± 5.26 BPRS-T: 26.9 ± 15.4
Simsek (2015) [46]	Europe	6 months	DSM IV	52	FE HC	14.6 ± 1.6 14.6 ± 1.6	42.3% 38.5%	---	100%	Serum	FE: 2.0 ± 1.9 HC: 3.4 ± 3.0	PANSS-P: 20.6 ± 7.4 PANSS-N: 29.0 ± 9.9 PANSS-GP: 33.9 ± 5.7
Song (2014) [47]	Asia	---	DSM IV	76	DN FE HC	22.54 ± 5.79 24.33 ± 7.42	61% 57%	---	100%	Serum	FE: 746.31 ± 171.89 (pg/mL) HC: 828.67 ± 109.30 (pg/mL)	PANSS-P: 21.85 ± 3.17 PANSS-N: 18.13 ± 2.28 PANSS-GP: 38.78 ± 2.30 PANSS-T: 78.72 ± 4.74
Sotiropoulou (2013) [48]	Europe	---	ICD-10	100	FE HC	29.84 ± 8.20 31.36 ± 7.96	68% 68%	---	100%	Serum	FE: 12.62 ± 1.86 HC: 14.52 ± 2.18	---
Theleritis (2014) [49]	Europe	---	ICD-10	239	FE HC	30.6 ± 9.3 32.1 ± 11.6	65.6% 48.7%	42.5% 59.2%	25.3%	Plasma	FE: 26104.6 ± 7202.7 (pg/L) HC: 24311.8 ± 6626.9 (pg/L)	---
Xiao (2017) [50]	Asia	---	DSM V	113	DN FE HC	25.0 ± 5.9 26.5 ± 6.6	53.4% 60%	---	100%	Serum	FE: 10.0 ± 2.8 HC: 12.3 ± 1.6	PANSS-P: 24.9 ± 5.9 PANSS-N: 19.6 ± 5.6 PANSS-GP: 30.9 ± 5.6 PANSS-T: 75.3 ± 8.0
Zakharyan (2014) [51]	Europe	---	DSM IV	208	FE HC	46.0 ± 9.8 37.3 ± 11.3	51.5% 48.6%	---	24.3%	Plasma	DN FE: 175.9 ± 13.5 HC: 245.3 ± 30.4	---

Abbreviations: DSM: Diagnostic and Statistical Manual of Mental Disorders; ICD: International Statistical Classification of Diseases and Related Health Problems; FE: First Episode, DN: Drug Naïve; HC: Healthy Control; BL: Baseline; FU: Follow up; Dx: Diagnosis; SD: Standard Deviation; Cauc: Caucasian, BDNF: Brain-Derived Neurotrophic Factor; PANSS-P: Positive and Negative Syndrome Scale-Positive; PANSS-N: Positive and Negative Syndrome Scale-Negative, PANSS-GP: Positive and Negative Syndrome Scale-General Psychopathology; PANSS-T: Positive and Negative Syndrome Scale-Total; BPRS-T: Brief Psychiatric Rating Scale-Total; BPRS-P: Brief Psychiatric Rating Scale-Positive; BPRS-N: Brief Psychiatric Rating Scale-Negative; SAPS: Scale for the Assessment of Positive Symptoms; SANS: Scale for the Assessment of Negative Symptoms.

**Table 2 brainsci-12-00414-t002:** Results of meta-regression on the difference in BDNF levels between First Episode Psychotic Patients and Healthy Controls at baseline.

Moderator	Beta	Standard Error	Z Value	*p* Value
Total N	0.002	0.001	2.027	0.043
Patient mean age	0.046	0.011	4.087	<0.001
% male	0.016	0.004	3.622	<0.001
% drug naïve patients	−0.009	0.002	−6.020	<0.001
Publication year	−0.012	0.009	−1.360	0.174
Duration of untreated psychosis (DUP)	0.0003	0.003	0.109	0.913
Baseline symptom severity	−0.005	0.003	−1.784	0.074
% white	−0.00002	0.001	−0.019	0.985

## Data Availability

Data used in the meta-analysis is available upon request.

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
