# Peer review of "Peripheral Blood Levels of Brain-Derived Neurotrophic Factor in Patients with First Episode Psychosis: A Systematic Review and Meta-Analysis"

_brainsci, 2022, doi:10.3390/brainsci12040414_

Round 1

Reviewer 1 Report

Manuscript number: brainsci-1596980

Manuscript title: Peripheral Blood Levels of Brain Derived Neurotrophic Factor in Patients with First Episode Psychosis: A Systematic Review and Meta-Analysis

Dear authors/editor

This is a fruitful and well-written meta-analysis following a systematic review. Indeed, the study included two meta-analyses. Authors examined relevant literature and sought an answer to the questions of “Is BDNF level different in FEP patients compared to the healthy population?” and “Does BDNF level alter with AP medication?” They found that BDNF, a well-recognized diagnostic and prognostic marker for schizophrenia, is lower in patients with FEP compared to controls, and interestingly, does not change with medication. Moderator analysis indicated that studies that included younger patients, higher symptom severity and more drug naïve patients had larger ES when compared to pooled ES of the whole analysis.

Strengths of the study

-           Publication bias was appropriately examined.

-           Literature search is adequately explained and PRISMA protocol was carefully followed.

-           Flowchart and tables are well-presented.

Limitations of the study

-           There is not given any limitation. I believe authors should discuss the generalizability of the results and other limitations.

Author Response

            Thank you for the suggestion. We added a paragraph to describe the limitations of the study. Please see Lines 275-283.

Reviewer 2 Report

This systematic review and meta-analysis (SRMA) performed by Singh et al. investigates peripheral blood levels of BDNF in patients with first episode psychosis. The paper is interesting and written well. Substantial amount of work has been put in. However there are several points I am about to make which questions the validity and proper conduct of a SRMA, and need  to revise.

As part of the Preferred Reporting Items for Systematic Reviews and Meta-Analyses (PRISMA), it is standard to register their systematic review protocol a priori, and at least before data extraction for transparency. These days, it is common practice to register on PROSPERO https://www.crd.york.ac.uk/prospero/

As this was not perform, it may limit the validity of the findings. Alternatively, the authors could have performed a scoping review (which does not aim to answer a specific research question) which does not require a protocol registered a priori.

These are two dire points. Unfortunately, failure to perform these cannot be undone. They had to be performed before proceeding with the SRMA.

No search string was provided by the authors. This could also not be found in the supplementary material. In adherence to PRISMA, this should have been provided otherwise this project is not repeatable and reproducible. We would not be able to also ascertain whether or not this search was carried out systematically under standardised protocols.

Additionally, “Literature search was conducted up to June 2021”, till which date exactly?

In Figure 1 of the PRISMA flow chart, the 1361 ‘irrelevant articles’ removed at the first stage of screening, please may we have a proper breakdown, as the one provided in the second stage of screening. ‘Irrelevant articles’ is too broad and I doubt proper screening may have been carried out without a fixed definition and record keeping.

In the introduction, the authors state the cohort of subjects were inconsistent, for example, some studies used first episode patients but others used chronic patients, which led to inconsistent findings in literature.

 “BDNF could serve as a potential diagnostic and prognostic biomarker, which may allow us to detect schizophrenia early and track treatment response. However, available 50 data on peripheral levels of BDNF in psychotic patients compared to healthy controls is not consistent[19]. For example, two studies showed a decrease of BDNF in a sample of psychotic patients compared to healthy controls[20,21], while other studies showed no statistical difference between patients and healthy controls[22,23]. Yet, some studies have 54 also shown that peripheral BDNF levels are increased in patients versus healthy controls[24]. One potential cause of the inconsistency is that some studies used first episode patients but others used chronic patients, Previous research has investigated the effect of antipsychotic treatment on peripheral BDNF levels in longitudinal studies, but the data is also inconsistent.”

How then, did the authors regulate and choose their exposed group? Such information was not provided. It seems that the exposed group were just pooled into one. A meta-analysis can only be as useful and strong as its primary studies. I would strongly suggest that the authors rectifiy this.

“If high heterogeneity was found, moderator analysis and meta-regression were conducted to examine the sources of heterogeneity.”

  • was this indeed performed?

Why was the Hedges’ G used as the effect size measure and not typically used measures such as the MD or SMD which may also answer the research question.

The authors stated one of the aim of the paper was to also examine studies that provided longitudinal data after the start of antipsychotic treatment to assess the validity of BDNF being a prognostic tool, how was this prognosis actually investigated with the current methods?

Reviewer 3 Report

Thank you for the opportunity to review the manuscript entitled "Peripheral Blood Levels of Brain Derived Neurotrophic Factor in Patients with First Episode Psychosis: A Systematic Review and Meta-Analysis” for Brain Sciences.  The core findings were that Peripheral BDNF level 22 was decreased in FEP compared to HC, but it did not change after treatment. These findings are meaningful and represent an important area of inquiry that is relevant to the readership of this journal. However, additional details are needed regarding the methods and results sections to better situate this study in the existing literature. In summary, the findings of the study are meaningful but more detail is needed in the manuscript overall. Please see my comments below:

  • Please, add Web Of Science in line 14.

  • Line 82: Please, report the exact search strategy carried out in Pubmed.

  • If authors provide PRISMA checklist as a supplementary data, the paper improve substantially.

  • The statistical analyses were well designed, although why the authors omitted tables/figures of mediaton and meta-regression analysis?

  • Visual inspection of the forest plots suggests the presence of influential studies (eg, Zakaharyan et al. (2014), Figure 2). Therefore, it is mandatory to perform an influence analysis.

  • Line 154: -0.89 to -0.37.

  • Line 182: Please, report results from Trim & Fill analysis.

  • Egger’s regression test detected publication bias more frequently than other tests (like Begg’s test) in meta-analyses of non-binary outcomes (Lin et al., 2018). Please, report Egger’s test p-value to assess publication bias. t would be interesting if the authors also reported fail-safe N.

Reference: Lin L, Chu H, Murad MH, Hong C, Qu Z, Cole SR, Chen Y. Empirical Comparison of Publication Bias Tests in Meta-Analysis. J Gen Intern Med. 2018 Aug;33(8):1260-1267. doi: 10.1007/s11606-018-4425-7.

Round 2

Reviewer 2 Report

The revisions and explanations provided were well deliberated and well executed.